# Coverage of completion of four ANC visits based on recommended time schedule in Northern Ethiopia: A community-based cross-sectional study design

Mulu Ftwi[1]*, Gebremedhin Gebre-egziabher Gebretsadik[1], Haftu Berhe[2], Mebrahtom Haftu[2], Gebrehiwot Gebremariam[3], Yemane Berhane Tesfau[4]

1 Department of Midwifery, College of Health Sciences, Adigrat University, Adigrat, Ethiopia, 2 School of Nursing, College of Health Sciences, Mekelle University, Mekelle, Ethiopia, 3 Department of Public Health, College of Health Sciences, Adigrat University, Adigrat, Ethiopia, 4 Department of Nursing, College of Health Sciences, Adigrat University, Adigrat, Ethiopia

* muluf99@gmail.com

## Abstract

### Background

Despite decades of implementation of maternal health care programs, the uptake of antenatal care services based on the recommended gestational age continues to be below the national and regional targets. Thus, this study aimed to assess the prevalence and factors related to the completion of four antenatal care visits among mothers who gave birth 6 months preceding the study.

### Method

We conducted a community-based cross-sectional study using both quantitative and qualitative approaches. The quantitative component included administering a pre-tested structured questionnaire to 466 mothers who gave birth 6 months preceding the study using a simple random sampling technique from respective Tabias. The quantitative result was analyzed using SPSS version 22. Bivariate and multivariate analysis was done to determine the association between independent and dependent variables. Variables were declared as statistically significant at P ≤ 0.05 in multivariable logistic regression model. The qualitative interview data were collected from eight mothers and four key informants recruited through purposive sampling method.

### Results

The overall prevalence of completion of four ANC visits based on the recommended time schedule was 9.9% (95% CI, 7.1–12.4). However, 63.9% of the participants attended four visits or more regardless of the recommended time schedule. Being member of community health insurance (AOR 2.140, 95% CI, 1.032–4.436), walking on foot less than or equal one hour to reach the health facility (AOR 3.921, 95% CI, 1.915–8.031), having workload at

**Data Availability Statement:** All relevant data are within the manuscript and its Supporting Information files.

**Funding:** We are very grateful to Mekelle University for providing financial support for this study. The funder had no role in study design, data collection and analysis, decision to publish, or preparation of the manuscript.

**Competing interests:** The authors have declared that no competing interests exist.

home (AOR 0.369, 95% CI, 0.182–0.751), and husband supported during antenatal care (AOR 2.561, 95% CI, 1.252–5.240) were independently associated with the completion of four ANC visits based on the recommended time schedule in multivariable analysis.

## Conclusion

The completion of four ANC contacts based on the recommended time schedule remains low in rural areas of Northern Ethiopia. Being a member of community health insurance, distance to the health facility, workload, and male involvement were associated with the completion of four ANC visits based on the recommended time schedule. The existing health system should consider improving the recommended ANC visits by integrating Community based interventions.

## Introduction

"According to WHO, Antenatal care (ANC) can be defined as the care provided by skilled health care providers to a woman from the beginning of pregnancy until the onset of labor It is a key entry point for pregnant women in order to receive a broad range of health promotion and preventive services, including screening for danger signs during pregnancy, delivery and the postpartum period [1–3].

In 2003, the World Health Organization (WHO) recommended that in low-income countries pregnant women without any complication should visit the health facility at least four times for ANC based on recommended schedules in their pregnancy period [3]. This was called Focused Antenatal care (FANC) model. The aim was to address evidence-based clinical interventions, counselling on maternal health, on birth and emergency preparedness [4]. Even though, it is reported that 83% of pregnant women received ANC at least once in their pregnancy period globally [5], some studies suggested that in low-income countries pregnant women often do not receive the recommended number of ANC visits [3]. For example, the report on sub-Saharan Africa pointed that 71% of pregnant women attend formal ANC at least once; however, only 44% attended ANC four or more times [5]. Similarly, a review of trends over the last two decades has shown high variability in the proportion of pregnant women making at least four ANC visits across the globe [6].

In 2016, the WHO reviewed the previous guideline, recommending at least eight ANC visits throughout the pregnancy as a means to decrease perinatal death and enhance maternal satisfaction, irrespective of the resource setting, compared to the FANC model. However, in the real world scenario, pregnant women are not even adhering to the FANC model. Thus, the first step could be trying to reach the four ANC visits goal.

In Ethiopia, though ANC services are provided in many places (community health post, health centres and hospitals), only 27% of rural women have four or more ANC visits [7]. In previous studies, level of education, ANC knowledge, service satisfaction, maternal age, parity and correct communication of pregnancy timing were found to be contributing factors for the completion of four ANC visits [8]. Women who have inadequate number ANC visits or start ANC later than the first trimester seem to have higher rates of poor pregnancy outcomes, such as low birth weight and pre-term birth. Besides, completing four ANC visits is pivotal in fostering, safe delivery, and postnatal care attendance among women in low-income countries [1]. Though information on completion of four ANC based on recommended time schedule is

crucial, to our knowledge there is dearth of information so far in Northern Ethiopia. Therefore, the aim of this study was to assess the prevalence and factors related to the completion of four ANC visit based on the recommended time schedule, in rural areas of Northern Ethiopia.

## Methods and materials

### Study area and period

The study was conducted in the Eastern zone, one of seven zones of Tigrai Region, Northern Ethiopia. Based on the 2007 Census, the total population of the zone was 850,000 with the annual expected number of deliveries estimated to be around 26,350 [9]. According to the 2007 (EFY) survey, the health service coverage in the eastern zone was approximately 85%. Eastern Zone has two general hospitals, five primary hospitals, 36 health centres, and 116 health posts [9]. The data were collected from March 9 to May 30, 2019, in the rural Eastern zone, Tigrai Region, Northern Ethiopia.

### Study design

A community-based cross-sectional study design was conducted. All mothers who gave birth within the six months preceding the data collection period were included in a quantitative study. In addition, Mothers who gave birth six months preceding the study period, midwife professionals, health extension workers, and Women's Development Army Members were purposively selected. To complement the quantitative evidence, we collected qualitative data from mothers other than who participated in the quantitative study.

### Source population

All mothers living in the Eastern Zone of rural Tigrai who gave birth six months preceding the study.

### Study population

Mothers who gave birth six months preceding the study period in the selected "Tabia's" of Eastern Zone of rural Tigrai region

**Study subjects for quantitative study.**   Those mothers who gave birth six months preceding the survey and selected for interview in the selected "Tabias" of Eastern Zone.

**Study subjects for qualitative study.**   Mothers, women who were members of the women's development armed, Midwifes and Health extension workers that purposively selected for interview in the selected "Tabias" of Eastern Zone.

### Sample size determination and sampling procedure

The sample size for the quantitative study was determined by the single population proportion formula, using the proportion of the completion of four ANC visits in rural women (27%) [10]. Therefore the total sample size was calculated with the assumption of 0.05 marginal errors and 95% confidence interval. Based on these assumptions, the sample size was estimated at 303. After considering 1.5 design effects and adding 5% for non-respondents the final sample size was taken as 477. Out of the seven woredas in the rural Eastern zone of Tigrai, three woredas were selected randomly and twenty-one "Tabias" were selected from the three woreda using simple random sampling to recruit the study participants. Finally, participants were selected using simple random sampling from the registration books for women who gave birth 6 months preceding the study in the selected "Tabias". For the qualitative part, eight mothers in the in-depth interview and four key informant interviews from women development group

and health extension workers were involved. The number of participants and time spent was determined based on the data saturation level.

## Operational definition

Completion of four ANC visits: Mothers were considered compliant when they attended four ANC visits based on the WHO recommended ANC schedules (1$^{st}$ visit at less than 16 weeks of gestational age, 2$^{nd}$ visit between 24th -28th week gestation, 3$^{rd}$ visit between 30th-32nd weeks gestation, and 4$^{th}$ visit at 36$^{th}$ week gestation or later) [11].

Components of ANC services: We conducted a composite score by giving minimum value of zero and maximum value one for each item. The component of ANC service was dichotomized as "yes" when participants scored above mean score value. Moreover, was categorized as "no" when participants scored below mean score value [4, 12, 13].

## Data collection procedures and tools

**Quantitative study.**   The data were collected using a pre-tested structured questionnaire. It was prepared in English and translated in to local language (Tigrigna) and then translated back to English to check its consistency by language experts. Twelve data collectors and four supervisors were recruited for the data collection.

**Qualitative approach.**   The interviews were conducted using an open-ended interview guide by the principal investigator. The data were also recorded digitally.

## Data processing & analysis

**Quantitative.**   After each questionnaire was checked for its completeness and consistency, data were coded and entered into SPSS™ version 22 for analysis purposes. Frequency tabulations were performed to analyse the distribution of factors among the women included in this study. Bivariate logistic regression analyses were conducted to assess the crude odds ratio (OR) between each of the factors and the completion of four antenatal visits. In the bivariate analysis model, each variable with p<0.2 was considered as a significant variable and included in the multivariate logistic regression model. Each variable with p ≤ 0.05 was considered as a significant variable in the final model. Multi-colinearity was checked among the independent variables using the variance inflation factor and the value was VIF = 1.086. The necessary assumption of model fitness during logistic regression was checked using the Hosmer and Lemeshow goodness of fit test statistics and the value was p = 0.410. Measures of association were calculated using adjusted odds ratio (AOR) with 95% confidence interval. Finally, variables p-value less than 5% of the adjusted odds ratio were declared as significant and presented using tables, graphs, and text.

## Qualitative approach

The qualitative component of the research were analysed after the analysis of the quantitative data. The interviews were recorded digitally, transcribed verbatim into the local language, and then translated into English by language experts, subsequently analyzed thematically using Atlas Ti™ version 7. Finally, barriers related to categories like the accessibility of transport, household workload, and distance from home to health facilities, as well as facilitator related categories such as health insurance and husband's involvement were identified.

Ultimately, the quantitative and qualitative data were triangulated at the results and discussion to improve the validity as well as to complement the quantitative findings.

## Data quality assurance

Data collectors, along with the supervisors, were trained regarding the purpose of the study, data collection procedures, study tools, and handling of collecting data. During the data collection period, close supervision was made daily by the supervisors.

## Ethical consideration

The study received ethical approval from the Institutional Review Board of the College of Health Sciences at Mekelle University (ERC 1258/2019). Written permission was taken from the Tigrai Regional Health Bureau and from the respective district administrations. Verbal permission was considered from "Tabia" administrators for facilitating the data collection. Besides, consent was secured from the respondents after explaining the purpose of the data collection. All information obtained from the participants was strictly kept confidential and were only being used for the purpose of the research.

## Results

### Socio-demographic and economic characteristics of respondents

A total of 466 mothers participated in this study with a response rate of 97.7%. The median age of participants were 27 years (±12) and an interquartile range of 22 to 34 years. Most of the study participants (n = 423, 90.8%) were Christian Orthodox. Majority, (n = 366, 78.5%) of the mothers were married and around two third of families earned (n = 385, 82.65%) with two thirds earning less than $54 USD per month. Of the study participants, 264 (56.7%) walked more than 1hr on foot to reach the health facility (Table 1).

### Obstetric history of study participants in the eastern zone of Tigrai region

Out of the total respondents, 257 (55.2%) were Para one and two. Above half, 250 (53.6%) of mothers give their first birth after their 20 years old. Regarding participants' birth intervals, 154 (49%) of the participants' current child was at a 3–5 year interval of their previous child. The majority (76.2%) of respondents reported their recent pregnancy was planned (Table 2).

### Source of information about ANC visit

Regarding to information about ANC visit, 465 (99.8%) of the respondent had an access to information. Majority of the mothers' source of information 368 (79%) were, from Health Extension Workers (Table 3).

### Uptake of antenatal care visits

The overall ANC attendance for at least one visit was found to be 98.1%. Of 298 (63.9%) mothers attended four and above ANC visits regardless of the recommended time schedule. Three hundred seventy-six (80.7%) of them were attended at health centres. Regarding the time of visits, 228 (48.9%) attended their first visit before 16 weeks of gestational age, 257 (58.4%) attended their second visits in between 24–28 weeks of gestational age, 178 (42.2%) attended their third visits from 30–32 weeks of gestational age, and 171 (58.%) completed their fourth visit after 36 weeks gestation. The overall prevalence of completion of four ANC visits based on the recommended schedule in our study was 9.9% (Table 4).

**Table 1. Socio-demographic characteristics of delivered mothers in the eastern zone of rural Tigrai Region, Northern Ethiopia 2019 (n = 466).**

| Variables Category | | Frequency (%) | Four ANC completion | |
|---|---|---|---|---|
| | | | Yes | No |
| Maternal age | <20 | 74 (15.9) | 3 (4.05%) | 71 (95.94%) |
| | 20–34 | 286 (61.4) | 30 (10.5%) | 256 (89.5%) |
| | 35–49 | 106 (22.7) | 13 (12.2%) | 93 (87.7%) |
| Maternal occupation | Housewife | 385 (82.6) | 46 (11.9%) | 339 (88%) |
| | Employed* | 81 (17.4) | 0 (0%) | 81 (100%) |
| Maternal education | No formal education | 138 (29.7) | 6 (4.3%) | 132 (95.7%) |
| | Elementary school | 161(34.5) | 19 (11.3%) | 142 (88.6%) |
| | Secondary school & above | 167(35.8) | 21(13%) | 146 (86.9%) |
| Paternal level of education | No formal education | 100 (21.5) | 6 (9%) | 94 (91%) |
| | Elementary School | 164 (35.2) | 18(10.97%) | 146 (89%) |
| | Secondary school & above | 202 (43.3) | 19 (9.4%) | 183 (90.5%) |
| Marital status | Married | 366 (78.5) | 40 (10.9%) | 326 (89%) |
| | Unmarried | 36 (7.7) | 3 (8.3%) | 33 (91.6%) |
| | Divorced | 18 (3.9) | 1 (5.5%) | 17 (94.4%) |
| | Widowed | 14 (93.0) | 0 (0%) | 14 (100%) |
| | Separated | 32 (6.9) | 2 (6.25%) | 30 (93.75%) |
| Distance from home to health care | >1hrwalk | 264 (56.7) | 14 (5.3%) | 250 (94.7%) |
| | ≤1hr walk | 202 (43.3) | 32 (15.8%) | 170 (84.2%) |
| Household size | <5 | 242 (51.9) | 18 (7.4%) | 224 (92.5%) |
| | ≥5 | 224 (48.1) | 28 (12.5%) | 196 (87.5%) |
| Family monthly income (USD) | <54 | 307 (65.9) | 25 (8.14%) | 282 (91.8%) |
| | 54–107 | 90 (19.3) | 12 (13.3%) | 78 (86.6%) |
| | >107 | 69 (14.8) | 9 (13%) | 60 (86.9%) |

*Employed = private or governmental employed

**Community participation related variables.** In our study, 251 (53.9%) of mothers were getting support from their husbands to attend the ANC visits. Two hundred forty two (51.9%) of study participants reported having health insurance. However, 191 (41%) of mothers attended pregnant women's forum (Table 5).

**Table 2. Obstetric history among delivered mothers in the eastern zone of rural Tigrai Northern Ethiopia 2019.**

| Variable | | Frequency | Four ANC completion | |
|---|---|---|---|---|
| | Category | (N, %) | Yes | No |
| Parity (n = 466) | 1–2 | 257 (55.2) | 25 (9.7%) | 232 (90.7%) |
| | 3–4 | 114 (24.5) | 9 (7.9%) | 105 (92. %) |
| | ≥5 | 95 (20.4) | 12 (12.6%) | 83 (87.4%) |
| Age at first pregnancy (n = 466) | Under 20 | 216 (46.4) | 20 (9.25%) | 196 (90.7%) |
| | ≥ 20 | 250 (53.6) | 26 (10.4%) | 224 (89.6%) |
| Birth interval (n = 314) | <3 yr | 95 (30.3) | 10 (10.5%) | 85 (89.4%) |
| | 3–5 yr | 154 (49.0) | 20 (13%) | 134 (87.5%) |
| | >5 yr | 65 (20.7) | 6 (9.2%) | 59 (90.7%) |
| Planned pregnancy (n = 466) | Yes | 355 (76.2) | 38 (10.7%) | 317 (89.3%) |
| | No | 111 (23.8) | 8 (7.2%) | 103 (92.7%) |
| Complication history (N = 466) | Yes | 53 (11.4) | 6 (11.3) | 47 (88.7%) |
| | No | 413 (88.6) | 40 (9.68%) | 373 (90.3%) |

**Table 3. Source of information about delivered mothers about ANC visits in eastern zone of rural Tigrai Region, Northern Ethiopia 2019 (N = 466).**

| Variable | Category | Frequency (%) | Four ANC completion | |
|---|---|---|---|---|
| | | | Yes | No |
| Ever heard about the ANC | Yes | 465 (99.8) | 46 (9.9%) | 419 (90.1%) |
| | No | 1 (.2) | 0 (0%) | 1 (100%) |
| Source of information | | | | |
| Husband | Yes | 87 (18.7) | 11 (12.6%) | 76 (87.3%) |
| | No | 379 (81.3) | 35 (9.23%) | 344 (90.7%) |
| Family member | Yes | 121 (26.0) | 13 (10.7%) | 108 (89.2%) |
| | No | 345 (74.0) | 33 (9.56%) | 312 (90.4%) |
| Health extension worker | Yes | 368 (79.0) | 39 (10.6%) | 329 (89.4%) |
| | No | 98 (21.0) | 7 (7.1%) | 91 (92.8%) |
| Midwife or Nurse, Health officer or Doctor | Yes | 255 (54.7) | 23 (9%) | 232 (91%) |
| | No | 211 (45.3) | 23 (10.9%) | 188 (89%) |
| Religious leader | Yes | 13 (2.8) | 0 (0%) | 13 (100%) |
| | No | 453 (97.2) | 46 (10.1%) | 407 (89.8%) |
| TV or Radio | Yes | 158 (33.9) | 20 (12.6%) | 138 (87.3%) |
| | No | 308 (66.1) | 26 (8.44%) | 282 (91.5%) |
| Newspaper | Yes | 34 (7.3) | 0 (0%) | 34 (100%) |
| | No | 432 (92.7) | 46 (10.6%) | 386 (89.3%) |
| Others source of information | Yes | 67 (14.4) | 3 (4.5%) | 64 (95.5%) |
| | No | 399 (85.6) | 43 (10.7%) | 356 (89.2%) |

**Table 4. ANC visit status by gestational age among delivered mothers in eastern zone of rural Tigrai, Northern Ethiopia, 2019.**

| Variables | Category (Yes/No) | Frequency (%) | Completion of four ANC visits | |
|---|---|---|---|---|
| | | | Yes | No |
| Have you attended ANC for recent pregnancy (n = 466) | Yes | 457 (98.1) | 46(10.06%) | 411(89.9%) |
| | No | 9 (1.9) | 0(0%) | 9(100%) |
| No of ANC visits (n = 457) | Once | 18 (3.9) | 0(0%) | 18 (100%) |
| | 2–3 times | 141 (30.9) | 0(0%) | 141(100%) |
| | ≥4 | 298 (63.9) | 46(15.4%) | 252(84.5%) |
| GA at first visit (n = 457) | ≥16 weeks | 238 (51.1) | 0(0%) | 238(100%) |
| | <16 weeks | 228 (48.9) | 46(20.1%) | 182(79.8%) |
| GA at second visits (n = 440) | <23 weeks | 121 (27.5) | 4(3.3) | 58(96.7) |
| | 24–28 weeks | 257 (58.4) | 28(10.8%) | 229(89.1) |
| | ≥29 weeks | 62 (14.1) | 13(20.6%) | 108(79.03%) |
| GA at third visits (n = 422) | <29 weeks | 163 (38.6) | 6(3.68%) | 75(96.3%) |
| | 30–32 weeks | 178 (42.2) | 18(10.1%) | 160(89.8%) |
| | >32 weeks | 81 (19.2) | 21(25.9%) | 142(74%) |
| GA at the fourth visits (295) | <36 weeks | 124 (42.0) | 16(12.9%) | 155(87%) |
| | ≥36 weeks | 171 (58.0) | 18(10.5%) | 106(89.4%) |
| Completion of four ANC visit based on recommended time | No | 420(90.1) | | |
| | Yes | 46(9.9) | | |

*GA: gestational age.

**Table 5. Community participation related variables of delivered mothers in the eastern zone of rural Tigrai, Northern Ethiopia, 2019 (N = 466).**

| Variable | Category (Yes/No) | Frequency (%) | Four ANC completion | |
|---|---|---|---|---|
| | | | Yes | No |
| Husband's support for ANC visits | Yes | 251 (53.9) | 33 (13.1%) | 218 (86.8%) |
| | No | 215 (46.1) | 13 (6%) | 202 (94%) |
| Mother's workload at home | Yes | 228 (48.9) | 14 (6.1%) | 214 (93.8%) |
| | No | 238 (51.1) | 32 (13.4%) | 206 (86.5%) |
| Member of women development group | No | 179 (38.4) | 17 (9.49%) | 162 (90.5%) |
| | Yes | 287 (61.6) | 29 (10.%) | 258 (90%) |
| Member of health insurance | Yes | 242 (51.9) | 33 (13.6%) | 209 (86.3) |
| | No | 224 (48.1) | 13 (5.8%) | 211 (94.1%) |
| Model family | Yes | 126 (27.0) | 10 (7.93%) | 116 (92%) |
| | No | 340 (73.0) | 36 (10.6%) | 304 (89.4%) |
| Pregnant women forum | Yes | 191 (41.0) | 22 (11.5%) | 169 (88.5%) |
| | No | 275 (59.0) | 24 (8.72%) | 251 (91.2%) |

## The components of ANC service given in the health facility

Based on literatures [4], we used nine components of ANC service to assess the compliance. In this study, majority 288 (61.8%) of mothers were received five and more than five components of the ANC service during their visits. Majority of 296 (63.5%) mothers were waiting for 30 and above minutes to receive for the ANC in the health facility. Around 163 (35%) of mother were not satisfied by the ANC service in the health facility (Table 6).

## Factors affecting the completion of four ANC visit

In this study, the distance from home to the health facility, husband's support during ANC visits, the workload at home, and health insurance membership were found significantly associated with the completion of four ANC visits in multivariable analysis.

**Table 6. The components of ANC service among delivered mothers in the eastern zone of rural Tigrai Northern Ethiopia, 2019.**

| Variables | Categories | Frequency | Precent |
|---|---|---|---|
| Give you iron tablet | Yes | 366 | 75.4 |
| | No | 100 | 24.6 |
| Take blood sample | Yes | 373 | 80 |
| | No | 93 | 20 |
| Take urine sample | Yes | 368 | 79 |
| | No | 98 | 21 |
| Counseling and HIV testing | Yes | 390 | 83.7 |
| | No | 76 | 16.3 |
| Counseling about nutrition | Yes | 361 | 77.4 |
| | No | 105 | 22.6 |
| Counseling about birth preparedness | Yes | 338 | 72.1 |
| | No | 128 | 27.9 |
| Counseling to complete four visits | Yes | 343 | 73.6 |
| | No | 123 | 26.4 |
| Waiting in the health facility | <30 minutes | 170 | 36.5 |
| | > = 30 minutes | 296 | 63.5 |
| Satisfied by the service | Yes | 303 | 65 |
| | No | 163 | 35 |

**Table 7. Factors affecting of participant with the completion of four ANC visit in the eastern zone of rural Tigrai, Northern Ethiopia, 2019.**

| Variables | Category | ANC completion | | COR(95% CI) | AOR(95% CI) |
|---|---|---|---|---|---|
| | | Yes (n = 46) | No (n = 420) | | |
| Age of mother | <20 years | 3 (6.5%) | 71 (16.9%) | 1 | 1 |
| | 20-34yrs | 30 (65.2%) | 256 (61%) | 2.773 (0.822–9.353) | 2.414 (0.689–8.460) |
| | 35-49yrs | 13 (28.3%) | 93 (22.1%) | 3.308 (0.908–12.05) | 3.366 (0.862–13.14) |
| Monthly income of the family | <1500 | 25 (54.3%) | 282 (67.1%) | 1 | 1 |
| | 1500–3000 | 12 (26.1%) | 78 (18.6%) | 1.735 (0.834–3.610 | 1.826 (0.821–4.062) |
| | >3000 | 9 (19.6%) | 60 (14.3%) | 1.692 (0.752–3.808) | 1.689 (0.698–4.092) |
| Distance home to the health facility | >1hr Walk | 14 (30.4%) | 250 (59.5%) | 1 | 1 |
| | < = 1hr Walk | 32 (69.6%) | 170 (40.5%) | 3.361 (1.74–6.48) | **3.94 (1.95–7.94)**\*\*\* |
| Mass media as us source of information | Yes | 20 (43.5%) | 138 (32.9%) | 1.572 (0.848–2.915) | 1.18 (0.592–2.350) |
| | No | 26 (56.5%) | 282 (67.1%) | 1 | 1 |
| Husband's support during ANC | Yes | 33 (71.7%) | 218 (51.9%) | 2.352 (1.204–4.595) | **2.56 (1.25–5.216)**\*\* |
| | No | 13 (28.3%) | 202 (48.1%) | 1 | 1 |
| Workload at home | Yes | 14 (30.4%) | 214 (51.9%) | 0.421 (0.218–0.18) | **0.375 (.186-.755)**\*\* |
| | No | 32 (69.6%) | 206 (49%) | 1 | 1 |
| Member of health insurance | Yes | 33 (71.7% | 209 (49.8%) | 2.563 (1.312–5.007) | **2.117 (1.031–4.346)**\* |
| | No | 13 (28.3%) | 211 (50.2%) | 1 | 1 |
| Components of ANC services | Yes | 33 (71.7%) | 255 (60.7%) | 1.643 (0.840–3.213) | 1.766 (0.857–3.640) |
| | No | 13 (28.3%) | 165 (39.3) | 1 | 1 |

\*Significant at p<0.05

\*\* significant at p<0.01

\*\*\* significant at p<0.00.

Women who had health insurance were 2.1 times more likely to complete four ANC visits as compared to non-insured women (AOR = 2.117, 95% CI:1.031–4.346) (Table 7). This finding was supplemented qualitatively: A male midwife with six years' works experience mentioned

*Since, those mothers who are a member of the health insurance, easily adapted the health facility's environment, they are frequently visit our hospital for ANC service compared to the mothers who are not. Not only during pregnancy, but also they receive services when they feel something immediately since no need of paying of pocket" at each service (KII, 4).*

Mothers walking on foot less than one hour to reach the nearest health facility were 3.9 times more likely complete ANC visits as compared to those who walked more than one hour (AOR = 3.939, 95% CI:1.953–7.943). This finding was also supported by all of the participants in the qualitative study. A mother said that *distance could be a main reason for attending the ANC visit on time. For example, the distance from my house to the health facility takes around two hours on foot; so many women, including me, are not visiting the health facility on time* (IDI, P5).

Mothers who were supported by their husbands during ANC visits were 2.6 times more likely to complete the fourth ANC visit at the recommended time as compared to women who had not supported by their husbands (AOR = 2.556, 95% CI: 1.309–5.552).

From the qualitative study, a Para 1, 22 year old mother stated

*Except few educated husbands most of them especially, in the rural area, do not know the importance of the ANC for the foetus and the mother. So, these educated husbands share workloads, give money for transport, support their wife to visit a health facility for ANC (IDI, 6).*

The odds of four ANC visit completion on mothers who had workload at home decreased by 62% compared to those who had no workload at home (AOR 0.375, 95% CI: 0.186–0.755) (Table 7). This result was also supplemented by the qualitative result: One health care provider mentioned that

*in our setting, females work equally with the males. Especially in the summer season, women participate in ploughing, weed, mowing collecting of grain, keeping of cattle, and preparing of food; this makes the pregnant mother too busy to come to our health facility for an ANC visit (KII, 4)*

## Discussion

We conducted a study on the prevalence and factors related to the completion of four ANC visits among women in a rural Northern Ethiopia. The prevalence of completion of four ANC visits was 9.9% (95% CI: 7.1–12.4). However, 63.9% attended four visits or more regardless the recommended time schedule. Distance from home to the health facility, husband's support for the ANC, workload at home, and being membership in a health insurance were determinants of completion of four ANC visits.

The WHO's ANC model recommends pregnant women should attend ANC with a minimum of four visits within the recommended gestational age [11].

Even though there is dearth of information about the prevalence of completion of four ANC visit based on the recommended time schedule, a single study conducted using DHS data in sub-Saharan Africa, Middle east, Asia, Latin America and Caribbean countries shows 14.9%-38.6%, 80.5%-89.0%, 44.0% -78.9%, and 54.4%-76.1% respectively [14]. This indicates that our finding is low compared to the aforementioned study. The reason might be the completion of the four ANC visit based on the recommended time schedule only consider at the first ANC visit, but there is no clear statement that shows either the remaining visits were based on the recommended time schedule or not. Besides, sample size difference might also a reason. Current studies in rural Tigrai also showed that Health Extension Workers are committed in providing ANC services at home [15]. Because of this mothers might become reluctant to the recommended schedule. In this study, 298 (63.9) % mothers visited four times and above for ANC service regardless of the recommended time schedule. This finding is similar to the annual report of Tigrai Regional health bureau (65%) and mini EDHS 2019 (63.9%) [16], however, slightly lower than a study conducted in Jimma town (77.9%) [17] and Dembecha (80%) [18]. This difference might be due to the fact that urban women are more accessible to health services than rural women. The finding of this study is higher than studies conducted in other parts of Ethiopia [19]. In this survey, regardless of the time schedule visit, there is higher ANC utilization. This might be, because, the Ethiopian ministry of health introduced the four ANC services indicators in the maternal health program so as to improve the number of visits and it indicates an improvement on the awareness of maternal and child health care services. In addition, the study area could be a reason for the different result. Similarly this result is higher than findings reported elsewhere in African countries [4, 20]. The possible reason could be due to time difference.

An estimated 49% of women had delayed first ANC visits in this study. This estimate reflects under-utilization of ANC and this could possibly contribute to high maternal mortality. This prevalence of late booking is lower than study conducted in Central Tigrai (85.6%) [21]. The difference was due to study area and the definition given for late booking. Our result was also lower than study conducted in studies conducted elsewhere in rural Ethiopia [19, 22]. This implies that a significant proportion of pregnant women did not initiate ANC visits at the recommended gestational age. In Ethiopia, ANC coverage increased from 34% in 2011 to 62% in 2016 [22]. This shift indicates the government is committed towards increasing the coverage but focus was not given to the timing of ANC. So, raising the ANC coverage only may not be adequate for the pregnant women who require the capacities and knowledge to utilize the service at the optimal time of gestational age. The first visit offers an opportunity to establish baseline information on the general well-being of the mother and the fetus.

Our study found that mothers who walk less than one hour to the nearest facility were 3.9 times more likely to complete four ANC visits than those who walk more than one hour. Studies conducted in Southern and South Western Ethiopia have reported similar findings [19, 23, 24]. This is because distance is considered the major factor that hinders service utilization, even though, other factors affect service utilization because, service utilization could also depend on the behaviour of the individuals [15]. It is the fact that accessibility of a health facility is the main barrier for maternal health service including ANC, especially in Sub-Saharan Africa, where Ethiopia is located [25].

Similarly, mothers who were supported by their husbands were 2.56 times more likely to complete four ANC visits. A similar finding was reported in Yem special woreda, [19] and Nigeria [26]. Besides, a systematic review and meta-analysis from low- and middle-income countries reported that male involvement in antenatal care had a positive impact on the uptake of maternal health [27]. Participants from qualitative component of this study also mentioned if the husband shared money and ideas with his wife during ANC service, the mothers experienced freedoms and fully enjoyed the ANC opportunities. Because, a husband has contributed significantly to women's decision to use reproductive health services, male involvement has a crucial contribution in enhancing the maternal and neonatal health service including ANC [28].

On the other hand, mothers who had a workload at home were 62% less likely to complete four visits than those mothers who had no workload at home. This finding could be because, pregnant mothers might give priorities to other routine activities like ploughing, mowing, keeping of cattle, preparing food and caring for the children, making timely visits for ANC service more difficult.

Mothers who had a membership in a community health insurance were 2.1 times more likely to complete the four ANC visits. This evidence was in line with a studies done in Nigeria [29], Manzi [30], and Ghana [31]. The possible explanation could be that even though ANC service is cost-free in Ethiopia, the health insurance increases the healthcare-seeking behaviour since it does not require any out of pocket payment and this idea also supported by the participants in the qualitative study. Evidences in rural Tigrai also showed that being member of community health insurance was associated with increased maternal service utilization [32]. Findings from Ghana, Indonesia and Rwanda reported that, community based health insurance have positive effect on the enhancement of the maternal health care utilization including ANC visit based on the WHO recommended time schedule [33].

## Limitation of the study

Recall bias might be present in the mother regarding the specific weeks of gestational age and ANC visits. Our participants did not exclude mothers who had a previous history of pregnancy

complications. Since we collected the data directly from the mothers, social desirability bias is possible.

## Conclusion and recommendation

Our findings show that the completion of four ANC visit based on the recommended time schedule remains low. Husband's involvement during ANC, and being a member of community health insurance, distance from home to health facilities and workload at home were variables associated with the completion of four ANC visit based on the recommended time schedule. The existing health system should consider improving the recommended ANC visit by integrating Community based interventions. Health policies should be strengthening partner involvement, especially in maternal health programs.

The Ethiopian government has worked a lot on the number of ANC visits regardless of the gestational age, but this is not a guarantee for increasing the maternal and neonatal health.

This is the first study that evaluates ANC visits in the context of the correct gestational age. The methods could be reproduced in other low-income countries settings or other places in Ethiopia.

## Supporting information

**S1 File. Dataset of the prevalence and factors related to the completion of four ANC visit using a community based cross-sectional.**
(SAV)

## Acknowledgments

We strongly acknowledge Eastern Zone of Tigrai Health and Administrative Offices for their continuous support during data collection. Our thanks goes to professor Pammla Petrucka for the grammatically edited of the manuscript. Finally, we are also grateful to the data collectors, supervisors and all study participants who participated in the study.

## Author Contributions

**Conceptualization:** Mulu Ftwi, Gebremedhin Gebre-egziabher Gebretsadik, Haftu Berhe, Mebrahtom Haftu, Yemane Berhane Tesfau.

**Data curation:** Mebrahtom Haftu, Yemane Berhane Tesfau.

**Formal analysis:** Mulu Ftwi, Gebremedhin Gebre-egziabher Gebretsadik, Haftu Berhe, Mebrahtom Haftu, Gebrehiwot Gebremariam.

**Investigation:** Mulu Ftwi, Haftu Berhe.

**Methodology:** Mulu Ftwi, Gebremedhin Gebre-egziabher Gebretsadik, Haftu Berhe, Mebrahtom Haftu, Gebrehiwot Gebremariam, Yemane Berhane Tesfau.

**Resources:** Mulu Ftwi.

**Software:** Mulu Ftwi.

**Supervision:** Mulu Ftwi, Mebrahtom Haftu.

**Validation:** Mulu Ftwi, Gebremedhin Gebre-egziabher Gebretsadik, Haftu Berhe, Mebrahtom Haftu, Gebrehiwot Gebremariam, Yemane Berhane Tesfau.

**Writing – original draft:** Mulu Ftwi, Gebremedhin Gebre-egziabher Gebretsadik, Haftu Berhe, Mebrahtom Haftu, Gebrehiwot Gebremariam, Yemane Berhane Tesfau.

**Writing – review & editing:** Mulu Ftwi, Gebremedhin Gebre-egziabher Gebretsadik, Gebrehiwot Gebremariam.

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
