## [Decision Letter · Decision Letter 0]

25 May 2020

PONE-D-20-10078

Prevalence and factors related to the completion of four antenatal care visits among mothers who gave birth 6 months preceding the study in the rural eastern zone of Tigrai, Ethiopia: A community-based cross-sectional study

PLOS ONE

Dear Dr. Beraki,

Thank you for submitting your manuscript to PLOS ONE. After careful consideration, we feel that it has merit but does not fully meet PLOS ONE’s publication criteria as it currently stands. Therefore, we invite you to submit a revised version of the manuscript that addresses the points raised during the review process.

We look forward to receiving your revised manuscript.

Kind regards,

Sara Ornaghi, M.D., Ph.D.

Academic Editor

PLOS ONE

Journal Requirements:

2. Please update both your Methods section and Ethics Statement to state whether informed participant consent was given, or a statement declaring that the institutional ethics board waived the need for consent. Please state what type of consent was given, i.e. written, verbal, etc.

3. We note that you have uploaded two tables both labelled table 6.  Please can you re-number your tables so that only one table is table 6. Please also ensure that you refer the renumbered Tables in your text; if accepted, production will need this reference to link the reader to the Table.

Reviewers' comments:

Reviewer's Responses to Questions

**Comments to the Author**

1. Is the manuscript technically sound, and do the data support the conclusions?

Reviewer #1: Yes

Reviewer #2: Partly

2. Has the statistical analysis been performed appropriately and rigorously? 

Reviewer #1: Yes

Reviewer #2: Yes

3. Have the authors made all data underlying the findings in their manuscript fully available?

Reviewer #1: Yes

Reviewer #2: No

4. Is the manuscript presented in an intelligible fashion and written in standard English?

Reviewer #1: No

Reviewer #2: No

5. Review Comments to the Author

Reviewer #1: The authors sought to perform a report on antenatal care completion in a rural area of Ethiopia. The study is well conducted, I would suggest the following revisions.

Abstract

- Page 1, line 30: In the section "results", I would add this concept after the first sentence: "63.9% attended four visits or more regardless the recommended time schedule".

- Page 1 line 37: Please correct the syntax: In summary, only one of ten women attended the four ANC contacts at the recommended gestational age. Being a member of health insurance, distance to the health facility, workload and male support were associated with the completion of four ANC visits.

- in the Keywords, I would substitute "recommended" with "recommended gestational age"

Introduction

I re-wrote the introduction from line 41 to 71, mainly for syntax and word choices.

"According to WHO, Antenatal Care (ANC) can be defined as the care provided by skilled health care providers to a woman from the beginning of pregnancy until the onset of labor (1). It is a key entry point for pregnant women in order to receive a broad range of health promotion and preventive services, including screening for complications during pregnancy, delivery and postpartum period (2).

In 2003 the World Health Organization (WHO) recommended that, in low-income countries, pregnant women without any complication should visit the health facility at least four times based on recommended schedules in their pregnancy period (3). This was called Focused Antenatal care (FANC) model.

The aim was to address evidence-based clinical interventions, counseling on maternal health, on birth and emergency management (4). Even though, globally, it is reported that 83% of pregnant women received ANC at least once in their pregnancy period (5), some studies suggested that in low-income countries settings pregnant women often do not receive the recommended number of ANC visits (3). For example, the report on Sub Saharan Africa pointed that 71% of pregnant women attended formal ANC at least once; however, only 44% attended ANC four or more times (5).

Similarly, a review of trends over the last two decades has shown high variability in the proportion of pregnant women making at least four ANC visits across the globe (6).

In 2016, the WHO reviewed the previous guideline, recommending at least eight ANC visits throughout the pregnancy as a means to decrease perinatal death and enhance maternal satisfaction, irrespective of the resource setting, compared to the FANC model. However, in the real world scenario, pregnant women are not even adhering to the FANC model. Thus, the first step could be trying to reach the four ANC visits goal.

In Ethiopia, although ANC services are provided in many places (community health post, health centers and hospitals), only 27% of rural women have four or more ANC visits (8). In previous studies, level of education, ANC knowledge, service satisfaction, maternal age, parity and correct communication of pregnancy timing were found to be contributing factors for the completion of four ANC visits (9). Women who have inadequate number of ANC visits or start ANC later than the first trimester seem to have higher rates of poor pregnancy outcomes, such as low birth weight and preterm birth. Completing four ANC visits is pivotal in fostering early (first trimester) involvement, safe delivery and postnatal care attendance among women in low-income countries (1)."

Study design

Page 2, Line 95: I would explain what is "social desirability bias". I think it is not obvious for an audience of obstetrics and gynecologists, maybe more easy to understand for psychologists.

Results

- Page 5, lines 175-177: please write n=423, 90.8%, n=443, 95.1% and n=366, 78.5%.

- Page 5, table 1: what does it mean "Maternal occupation: employed"? What type of job? Please specify.

- Page 9 table 5: how did you define the dichotomic variable: "mother's workload at home"? Please specify also here (it is better explained at page 11 lines 262-266).

Discussion

- I was impressed by this concept: the prevalence of women attending the four ANC visits at recommended gestational age (GA) is 9.9%. However, 63.9% attended four visits or more regardless the recommended time schedule. At a first glance one might think that the main issue is attending visits, regardless the GA. But then, I focused on the fact that the majority of population attends four visits, but the issue is the correct timing. Maybe health care providers should stress during the first visit the importance of the correct timing of obstetrical checks. I would relate this to two important variables in Table 6: 26.4% did not receive counseling about completing four visits and 63.5% waited more than 30 minutes in the health facility. I argue that an improvement of these two components of ANC services could lead to better adhesion to 4 ANC visits. Please expand on this.

- Page 12, Line 279- 283: remove these sentences.

- Page 12 line 284 "compared to the study performed in Jimma Town"

- Page 12, line 286: " This might be due to the difference..."

- page 13 line 301: what do you mean with the term "utilization"?

- Page 13 line 310: an estimated 49% of women who had delayed first ANC visits

- Page 14 line 316: Our result was also lower than...

- Page 14 lines 318-319 : did not initiate, please remove "were" and choose between "by" or "at"

- Page 14 line 343: this idea

Conclusions

- Page 15 line 346: what do you mean with "plus"?

- Page 15 line 352: I would substitute "reducing" with " increasing" and remove lines 353 and 354 because you have already expressed this concept.

Please also remove lines 360/361

- Page 15 lines 363-367: please rewrite: " This is the first study that evaluates ANC visits in the context of the correct gestational age. The methods could be reproduced in other low-income countries settings or other places in Ethiopia".

Limitations

Please remove lines 369-370: you already explained this concept.

Reviewer #2: Dear authors, I would like to thank for your concern on maternal health care service utilization .Your manuscript need major revision especially on grammatical arrangement through out the manuscript.Beside this ,the discussion part is not well organized ,coherent and well sounded. See all the comments bellow :

1. Line -37-40-the abstract session doesn’t show recommendation

2. Line 48 –new sentence should be started with capital letter after punctuation .

3. Line 66- ‘ health provider knowledge for early pregnancy months’ is not clear ,how could be measure it ?

4. Line 67-71

5. Line 76-

6. Line 80- “quantitative and descriptive qualitative approach methods…. “ is there such type of study design ? try to correct it…

7. Line95 –remove the following phrases ….‘for inclusion ‘ and complete social desirability as social desirability biases

8. Line 101 what is tabia ?is it kebele or Gotte?,you need to operationalize it or describe on the method part

9. Line 119 and 128-131 have certain grammatical error .

10. Line 164,replace the word ‘ethics ‘with that of ethical’

11. Line174 needs measurement ,

12. Line 176-180 -most of the figures listed on the table are repeated on the text .so , make it precise .

13. Line 181-try to correct the heading of table 1 .Study subjects were those mothers who already gave birth 6 month prior to the study .but you considered as t says delivering mother .So, see it again including other tables and figures accordingly.

14. 182-correct the word tigrai region

15. Table one- who is literate on your study? You need to put operational definition if there is new thing otherwise those who were attending primary and secondary school should have another meaning.

16. Line-185-189 needs grammatical arrangement

17. Table 2- what is your base to categories parity? Do you think there is no difference b/n Para one and Para two mothers on utilization and timely initiation of antenatal care ?

18. On the same table ( table 2) age of mothers at first pregnancy was categorized as <19 and >20 what about those mothers whose age was 19 and 20 ?

19. Line 193-197-keep its coherence

20. Table 4-correct the heading as other tables .women developmental army group leader should have 2 option s(yes /no )but here is it has 3 alternatives .s ee it carefully

21. Line 210-be consistent headings .

22. It is not related with your objective

23. How did you measured uptake of components of antenatal care service ? how much service is required to say ‘ yes’ in each visit ?N:B utilization of all component of ANC service can not affect ANC completion but the reveres might be true.

25. Line - 276 being membership in health insuranceis different from health insurance plan .DISCUSSION

26. LINE 2976-324 not in line with your objective .There is a variation b/n yours and others

27. Discussion should be in line with your 2 objectives ,precise and arranged coherently .But ,here it includes conclusion s and back ground information about maternal health .

28. Line 330-331-you are using similar citations( R-15) for a study conducted in Uganda and yem ,how could be ?

29. Line 334-338- try to relate your finding with others.

30. Line 343 replace’ …his idea… ‘with ‘…this idea…

31. Conclusion part Line 346’-no need of repetition of result here rather you can say high or low according to others result

32. 351-358 shows the significance of the study and Justification so ,remove it since it is not the right place

33. Line 360-363the recommendations ; the recommendations part deals about awareness creation availability and accessibility of ANC service However you didn’t show the gap on the result part .So , the recommendation should be in line with your findings which can answer your objectives.

34. Line 370in the limitation part recall bias was one of the problem you faced during data collection

How did you managed it?because it directly affect your outcome variable .

35. Line 371-372what was the need of excluding mothers who had a previous history of pregnancy complication ?Even it could be one of the factor to attained ANC timely .Idont think so ,it could be considered as limitation of study .

36. Reference –line 410,REF 13 and line 416 R16 are similar references

37. Line414 REF14 and REF17 are similar references.

6. PLOS authors have the option to publish the peer review history of their article (what does this mean?). If published, this will include your full peer review and any attached files.

Reviewer #1: Yes: Annalisa Inversetti

Reviewer #2: No

---

## [Author Response · Author response to Decision Letter 0]

28 Jun 2020

We will very happy if accepted the publication of the our manuscript.

---

## [Decision Letter · Decision Letter 1]

17 Jul 2020

Coverage of completion of four ANC visits based on recommended time schedule in Northern Ethiopia: a community-based cross-sectional study design

PONE-D-20-10078R1

Dear Dr. Beraki,

We’re pleased to inform you that your manuscript has been judged scientifically suitable for publication and will be formally accepted for publication once it meets all outstanding technical requirements.

Kind regards,

Sara Ornaghi, M.D., Ph.D.

Academic Editor

PLOS ONE

Additional Editor Comments (optional):

Reviewers' comments:

Reviewer's Responses to Questions

**Comments to the Author**

1. If the authors have adequately addressed your comments raised in a previous round of review and you feel that this manuscript is now acceptable for publication, you may indicate that here to bypass the “Comments to the Author” section, enter your conflict of interest statement in the “Confidential to Editor” section, and submit your "Accept" recommendation.

Reviewer #1: All comments have been addressed

2. Is the manuscript technically sound, and do the data support the conclusions?

Reviewer #1: Yes

3. Has the statistical analysis been performed appropriately and rigorously? 

Reviewer #1: Yes

4. Have the authors made all data underlying the findings in their manuscript fully available?

Reviewer #1: Yes

5. Is the manuscript presented in an intelligible fashion and written in standard English?

Reviewer #1: Yes

6. Review Comments to the Author

Reviewer #1: (No Response)

7. PLOS authors have the option to publish the peer review history of their article (what does this mean?). If published, this will include your full peer review and any attached files.

Reviewer #1: **Yes: **Annalisa Inversetti

---

## [Editor Report · Acceptance letter]

3 Aug 2020

PONE-D-20-10078R1 

Coverage of completion of four ANC visits based on recommended time schedule in Northern Ethiopia: a community-based cross-sectional study design 

Dear Dr. Ftwi:

I'm pleased to inform you that your manuscript has been deemed suitable for publication in PLOS ONE. Congratulations! Your manuscript is now with our production department. 

Kind regards, 

on behalf of

Dr. Sara Ornaghi 

Academic Editor

PLOS ONE